# Stable Magnetorheological Fluids Containing Bidisperse Fillers with Compact/Mesoporous Silica Coatings

**DOI:** 10.3390/ijms231911044

**Published:** 2022-09-20

**Authors:** Martin Cvek, Thaiskang Jamatia, Pavol Suly, Michal Urbanek, Rafael Torres-Mendieta

**Affiliations:** 1Centre of Polymer Systems, University Institute, Tomas Bata University in Zlín, Trida T. Bati 5678, 760 01 Zlín, Czech Republic; 2Institute for Nanomaterials, Advanced Technologies and Innovation, Technical University of Liberec, Studentska 1402/2, 461 17 Liberec, Czech Republic

**Keywords:** smart materials, magnetorheology, surface texture, mesoporous silica, sedimentation, suspensions, coating, nano-layer

## Abstract

A drawback of magnetorheological fluids is low kinetic stability, which severely limits their practical utilization. This paper describes the suppression of sedimentation through a combination of bidispersal and coating techniques. A magnetic, sub-micro additive was fabricated and sequentially coated with organosilanes. The first layer was represented by compact silica, while the outer layer consisted of mesoporous silica, obtained with the oil–water biphase stratification method. The success of the modification technique was evidenced with transmission electron microscopy, scanning electron microscopy/energy-dispersive X-ray spectroscopy and Fourier-transform infrared spectroscopy. The coating exceptionally increased the specific surface area, from 47 m^2^/g (neat particles) up to 312 m^2^/g, which when combined with lower density, resulted in remarkable improvement in the sedimentation profile. At this expense, the compact/mesoporous silica slightly diminished the magnetization of the particles, while the magnetorheological performance remained at an acceptable level, as evaluated with a modified version of the Cross model. Sedimentation curves were, for the first time in magnetorheology, modelled via a novel five-parameter equation (S-model) that showed a robust fitting capability. The sub-micro additive prevented the primary carbonyl iron particles from aggregation, which was projected into the improved sedimentation behavior (up to a six-fold reduction in the sedimentation rate). Detailed focus was also given to analyze the implications of the sub-micro additives and their surface texture on the overall behavior of the bidisperse magnetorheological fluids.

## 1. Introduction

A current trend is to pursue technological innovation in connection with “smart” materials able to adapt their physicochemical characteristics in response to various external stimuli, such as magnetic [1] or electric fields [2], mechanical strain [3], etc. A magnetorheological (MR) fluid constitutes a good example of such a material, which is defined as a mixture of magnetic micro-particles dispersed in an insulating carrier liquid. When exposed to an external magnetic field, they dramatically change their rheological, viscoelastic, thermal and acoustic properties due to the formation of inner particle column-like structures [4,5]. These changes occur rapidly (in milliseconds) and revert once the magnetic field is removed; such remarkable behavior has been extensively investigated for a variety of sectors including automotive [6], robotics [7], civil engineering [8] and biomedical [9].

Since fluids of this type are composed of materials with a huge density mismatch (almost an eight-fold difference), one can assume a serious sedimentation issue. This drawback has to be suppressed in order to develop MR fluids suitable for industrial applications [10]. Several approaches have been developed to enhance stability and thereby avoid gravitational settling, including: (i) encapsulating magnetic particles within organic/inorganic coatings, or even multi-shell layers, in order to reduce bulk density and heighten interactions with the liquid medium [11,12]; (ii) applying magnetic/non-magnetic nano-particles (NPs) that act as physical barriers to inhibit settling [13,14,15]; (iii) replacing the carrier medium with viscous liquids [16] or gel-like media [17]; and (iv) dispersing special types of particles, i.e., hollow/porous particles [18].

These approaches have met with various degrees of success. In particular, hollow fillers exhibit remarkable improvements in sedimentation stability due to their low density which comes from the cavity [18]. For example, Choi et al. [18] used distillation–precipitation actions to synthesize hollow polydivinylbenzene@Fe_3_O_4_ particles. The authors utilized silica (SiO_2_) particles to fabricate a sacrificial core, which was then etched using hydrofluoric acid (HF), giving rise to a hollow structure. In terms of magnetorheology, little attention has been paid to employing magnetic particles as cores with lightweight porous shells, preferably with tailored surface properties. In this context, Chuah et al. [19] recently fabricated carbonyl iron (CI) particles encased within a polystyrene (PS) foam layer. They carried out conventional dispersion polymerization to synthesize the CI@PS material, while the compact PS layer was further foamed with supercritical CO_2_ serving as a foaming agent. This approach, however, required special experimental apparatus, such as a complex pressurized reactor [19]. Therefore, there is a high demand for the development of easily accessible core/porous-shell particles for MR purposes. 

Magnetic core/porous-shell particles, particularly nano-sized, have been reported in various areas of research, e.g., catalysis, magnetic resonance imaging, electromagnetic shielding and bimolecular separations [20,21,22]. The porosity of the final structures in the referenced cases was induced either directly or after synthesizing a core–shell hybrid. In relation to organic shells, template materials are usually removed by calcination or selectively dissolved in organic solvents [23]. Out of various organic materials investigated, mesoporous silica has attracted special attention for its versatility, as it has the capacity to craft different nano-structures [23,24]; indeed, the pore size of the silica coating is easily controlled by external parameters such as reaction temperature, time, etc. It was reported that Wei et al. [25] fabricated magnetic NPs with hierarchical tunable mesopores and a high surface area through oil–water biphase stratification. They later extended the method [22] to synthesize “dual-pore” mesoporous silica nano-spheres, revealing the huge potential in the fine-tuning of the interfacial activity.

In contrast to published studies on compact coatings in connection with magnetorheology, this paper investigates the implications of encasing sub-micro NPs in mesoporous shells that further serve as additives, giving rise to bidisperse MR fluids. Produced with the biphase stratification method, the resulting magnetic NPs with the mesoporous silica coating (hereinafter referred to as “NPs@m-TEOS”) exhibited improved wettability and reduced density, thereby substantially counteracting the sedimentation instability associated with conventional MR fluids. To our knowledge, the NPs@m-TEOS described herein possessed the highest ever specific surface area (SSA) reported for magnetic additives applied in MR fluids. The experimental design additionally permitted the comparison of the implications of different shell types, i.e., compact vs. mesoporous, revealing the relevance of surface texture in research on innovative MR fluids.

## 2. Results and Discussion

The size and morphology of the samples was investigated with SEM and TEM, as depicted in Figure 1. The neat NPs possessed a spherical shape and great uniformity; the size distribution histogram fitted with the Gaussian function showed a mean particle diameter of 248 ± 54 nm (*R*^2^ = 0.91). The corresponding TEM detected that each particle was formed as an assembly of primary NPs, which appeared as slight wrinkles at the edges of the coarse NPs. The NPs@TEOS had a mean particle diameter of 255 ± 46 nm (*R*^2^ = 0.95) and a smooth, compact TEOS coating with an approximate thickness of 40–50 nm (depending on particle size). Regarding the NPs@m-TEOS, the additional layer with a thickness of 20–30 nm was clearly distinguishable through difference in contrast. As a result, the average particle size increased to 296 ± 51 nm (*R*^2^ = 0.82), however, a certain role had also played the cross-coating phenomenon [24]. The TEM images for the NPs@m-TEOS clearly presented dendritic radial mesoporous channels with open pores, indicating their high SSA and great potential for MR applications. The electron microscopy tests suggested that the biphase stratification method was successful.

EDX spectra were recorded to discern the elemental composition of the nano-powders, and to detect any impurities stemming from the precursors. As detailed in Figure 2, the neat NPs were composed of the expected elements, i.e., Fe and O. A small amount of C was attributed to the carbon tape that served as a support for mounting the samples. As anticipated, the TEOS and m-TEOS-coated NPs exhibited a remarkable peak at 1.74 keV related to the binding energy of Si. Although the EDX technique is usually applied for qualitative analysis of materials, the NPs@m-TEOS exhibited somewhat stronger signals for silicon and oxygen (26.3 and 44.0 wt%), compared to those of the NPs@TEOS (15.9 and 32.9 wt%), indicating the presence of a thicker silica shell; this finding corresponded to the TEM observations (Figure 1b,c). To conclude, the EDX spectra evinced the success of the modification, and further proved that the samples were free of impurities.

Figure 3 shows the FTIR spectra for iron oxide NPs and their TEOS and m-TEOS-coated analogues, which were recorded to identify the surface functional groups in the samples. Characteristic peaks for iron oxides were reported in the literature beyond the investigated limit, typically at approximately 577 and 631 cm^−1^ in relation to Fe–O bonds [26]. Herein, the broad absorption peak at approximately 3423 cm^−1^ and the peak at 1606 cm^−1^ were assigned to the bending vibration of absorbed water and O–H stretching, respectively [27]. A peak at 1401 cm^−1^ was attributed to an O–H bond of ethylene glycol (served as a reaction medium), while a low intensity peak at 1069 cm^−1^ related to a C–C bond of the same organic compound. These results indicated that a trace amount of ethylene glycol was possibly present on the surfaces of the NPs [28]. The FTIR spectra for the TEOS-coated NPs, either with the compact coating or mesoporous variant, were almost identical due to the same surface chemistry. In detail, the significant absorbance peak located at 1069 cm^−1^ was assigned to the antisymmetric stretching of Si–O–Si groups, the small, sharp peak at 947 cm^−1^ corresponded to the vibration of Si–O chemical bonds, while that at 806 cm^−1^ denoted the presence of Si–C bonds, i.e., typical indications of organosilane compounds [29]. The FTIR spectra for the NP@TEOS and NPs@m-TEOS materials confirmed that the magnetic NPs were successfully coated.

Both the high SSA of the particles and their porous features were factors crucial to their wettability in the carrier medium [30]. In this context, nitrogen adsorption/desorption isotherms of the additives were obtained, and values calculated for the SSA with adherence to the BET theory, as detailed in Figure 4. The isotherms for the neat NPs and their TEOS-coated analogues were classified as reversible type-II, according to a IUPAC technical report [31], while NPs@m-TEOS resembled a type-IV isotherm [24]. All the isotherms exhibited negligible hysteresis, indicating the absence of interactions between the particles and the adsorbed molecules. The neat NPs exhibited an SSA of 47 m^2^/g, while this parameter notably decreased to 9.5 m^2^/g after applying the TEOS coating. This behavior was expected considering the uneven surface of the neat NPs in comparison to the compact TEOS coating, in agreement with the TEM observations (Figure 1). Upon formation of the mesoporous layer, the SSA value increased significantly to 312 m^2^/g, even exceeding SSA values typical for fumed silica [32]. In this context, it is notable that silica NPs are widely utilized as a stabilizer [33] and thixotropic agent [34] in MR fluids. Unlike commercial silica, the NPs@m-TEOS particles demonstrated a magnetic response (as shown below), which implied they played an active role in forming the particle structures under the magnetic field. Similar trends were observed for total pore volume, which decreased from 0.098 to 0.063 cm^3^/g after applying the compact TEOS coating, whereas 0.451 cm^3^/g was attained following the application of the mesoporous coating, signifying enhanced kinetic stability when in the bidisperse MR suspension. In connection with this, heightened chemical stability of the coated particles was anticipated as well as decreased abrasiveness due to the protective effect exerted by the organo-shell coatings [35].

Despite the importance of surface properties, SSA values for constituents of magnetic fluids have rarely been published. As seen in Table 1, the SSA of magnetic nano-fillers [30,36,37,38] usually ranges between 40 and 190 m^2^/g, whereas for non-magnetic nano-fillers such as silica [32], it attains values of approximately 115–200 m^2^/g. These values were exceeded by the NPs@m-TEOS particles (312 m^2^/g), revealing that the biphase stratification method had the potential to facilitate formation of an active interface that could be penetrated by a carrier liquid. Higher SSA values have been reported only for extremely small, non-magnetic NPs (>10 nm) [34,39].

The isothermal magnetic loops of the samples are displayed in Figure 5a. As can be seen, neither applying the TEOS coating, nor the formation of the mesoporous structure affected the intrinsic magnetic properties of the materials, and every VSM curve showed a typical sigmoidal shape, as observed for other magnetite-based materials, e.g., PS/Fe_3_O_4_ hybrids [41]. In order to determine the saturation magnetization (*M*_S_) of the NPs, the data was fitted by applying the Jiles–Atherton (J–A) model [25], that can be read as follows:(1)$$M\left(H\right)=M_{S}\left[\coth\left(\frac{H_{e}}{A}\right)-\left(\frac{A}{H_{e}}\right)\right]$$
where *M* denotes the magnetization, *H* is the magnetic field strength, *M*_S_ is the saturation magnetization and *A* is the parameter related to the magnetization shape without hysteresis, while *H*_e_ is the effective magnetic field strength, calculated as:(2)$$H_{e}=H+\alpha M$$
where *α* is the coefficient describing the coupling between domains. The J–A model agreed well with the data (*R*^2^ > 0.996), giving *M*_S_ values of 55.6, 34.1 and 23.7 emu/g for the neat NPs, NPs@TEOS and NPs@m-TEOS, respectively. A decrease in *M*_S_ values, proportional to the thickness of the silica (Figure 1b,c), was expected considering the non-magnetic property of the silica coatings. The hysteresis loops were narrow (Figure 5a, inset), and every sample had a coercivity of approximately 0.26 kA/m, which was supported by low values for the *α* parameter in the J–A model [42]. It is noteworthy that these NPs required relatively low magnetic fields (above 300 kA/m) to reach a close-to-saturation state, which is highly desirable for rapid field-on/off responses and reduced agglomeration [28]. These findings indicated that the formulated bidisperse MR suspensions were highly sensitive to external magnetic fields.

Additionally, the XRD pattern for the neat NPs (Figure 5b) was recorded to correlate magnetic properties with crystallographic structure. The positions of the peaks perfectly matched the standard (PDF card No. 00–019–0629) corresponding to cubic-phase magnetite, Fe_3_O_4_. The presence of γ-Fe_2_O_3_ could not be ruled out, however, since the *M*_S_ values for pure Fe_3_O_4_ are usually higher [11,28,43]. The mean crystallite size calculated from the parameters of the (311) diffraction plane by applying the Scherrer equation was ~8 nm. No additional diffraction lines were found in the spectrum, indicating that the sample was free of impurities. 

The carrier liquid was considered in relation to potential MR devices. Silicone oil was selected as a suitable medium for its thermal stability, wettability with silica-coated materials and relatively low price [29,44]. Besides the CI particles, nano-additives were dispersed in the silicone oil, giving rise to bidisperse MR suspensions. The amount of additives represented 2.5 wt% of the total suspension weight, which seemed sufficient for inducing enhancing effects [45]. Figure 6a illustrates the rheological response of the MR fluids during the ramp shear flow test. Interestingly, the bidisperse MR fluids exhibited slightly lower off-state data than the reference, implying that the NPs prevented the CI particles from aggregating, hence, facilitating such flow [46]. Upon exposure to a magnetic field, the samples exhibited greater shear stresses, which constituted an important phenomenon called the MR effect (example references [5,47]). This occurred through magneto-induced dipole–dipole interaction between the primary CI particles and additives, thereby forming a series of structured chains [11,45].

Under low magnetic fields (below 72 kA/m), the MR fluid supplemented with neat NPs exhibited higher shear stresses than the reference. Based on the empirical model reported by Anupama et al. [48], such rise in shear stress was initiated by a sharp increase in the magnetization of the NPs (Figure 6a), which in turn, supported interaction between the binary particles. Considering the dimensions of the NPs, the typical gap-filling effect was most likely limited; therefore, a lesser extent of enhancement was achieved than reported for nano-sized (<100 nm) additives [11]. The analogues supplemented with NPs@TEOS and NPs@m-TEOS demonstrated shear stress values comparable to the reference fluid. 

In fields of greater strength (above 144 kA/m), the bidisperse MR fluids showed slightly lower shear stress values than the reference. Possible factors responsible for this behavior included achieving the saturated state of the NPs, and limited void bridging for the NPs of such dimensions [48]. In order to evaluate differences in the stiffness of the structures in the various MR fluids, the flow curves were fitted by applying a suitable viscoplastic model. In light of recent advances in the parametric modelling of MR fluids [49], a modified Cross (mCR) model was selected as the most suitable option, as given below: (3)$$\tau =\tau_{0}+\left[\eta_{P}{\left(1+\alpha \dot{\gamma }\right)}^{-m}\right]\dot{\gamma }$$
where τ represents the shear stress, $\tau_{0}$ is the dynamic yield stress, $\dot{\gamma }$ denotes the shear rate, $\eta_{P}$ constitutes the plastic viscosity, α is the time constant and *m* stands for the dimensionless exponent. As illustrated in Figure 6b, the presence of sub-micro NPs led to a slight reduction in the dynamic yield stress under high magnetic fields, however, this drop was compensated by the lubricating effect of the additives in the off-state. For this reason, the presence of the additives resulted in comparable or even higher relative MR effects, as elaborated in the following section.

Testing under dynamic magnetic fields was performed to simulate the operation of real-world MR devices [50]. Figure 7 displays the results of the repeatability experiment under 216 kA/m, where the shear stress abruptly increased or decreased once exposed to the external magnetic field or after it was switched off, respectively. This phenomenon occurred within numerous loading cycles, evidencing the reversible formation of particle chains. It is noteworthy that the MR fluids supplemented with the NPs were able to stabilize the desired on-state values much faster than the reference fluid; in this context, better control over rheological properties is highly beneficial for practical applications. The performance of the MR fluids was compared next by applying the “MR effect” formalism as given by the following equation: (4)$$\mathrm{MR}\;\mathrm{effect}=\frac{\tau_{216\;\mathrm{kA}/m}-\tau_{\mathrm{off}}}{\tau_{\mathrm{off}}}\times 100\%$$
where $\tau_{\mathrm{off}}$ and $\tau_{216\;\mathrm{kA}/m}$ represent the average shear stress in the off-state and under a given magnetic field, respectively. The reference MR fluid attained the relative MR effect of 248 ± 8%, while an increase of up to 293 ± 14% was observed after bidispersing with the NPs@TEOS, which was attributed to a lower friction in the off-state conditions (Equation (4)). The MR fluids supplemented with the neat NPs and NPs@m-TEOS exhibited a slightly diminished yet acceptable MR effect, specifically, 229 ± 9% and 189 ± 10%. The results demonstrated that the magnetization of the NPs (Figure 6a) was not a single factor affecting the behavior of the bidisperse MR fluids, and the quality of the surface texture of the NPs was considered a highly relevant factor [11].

To better understand the behavior of the investigated MR fluids, viscosity was plotted against the dimensionless Mason number, *Mn*. The *Mn* groups together the magnetostatic and hydrodynamic forces as two dominant influences on an MR fluid under a steady flow [51], and is formulated as:(5)$$Mn=\frac{8\eta_{c}\dot{\gamma }}{\mu_{0}\mu_{c}\beta^{2}H^{2}}$$
where $\eta_{c}$ is the viscosity of the carrier liquid, $\dot{\gamma }$ is the shear rate, *H* is the magnetic field strength and β represents the magnetic contrast factor given by the equation below: (6)$$\beta =\frac{\mu_{p}-\mu_{c}}{\mu_{p}+2\mu_{c}}$$
where $\mu_{p}$ denotes the permeability of the particles and $\mu_{0}$($\mu_{c}$) is the permeability of free space (the MR fluid). The result is detailed in Figure 8, which compares the states of the reference MR fluid and its bidisperse analogue supplemented with NPs@TEOS. For the former of the two suspensions, the data taken under different magnetic fields collapsed into a single master curve, suggesting that magnetostatic and hydrodynamic forces dominated the problem, whereas other forces, such as the interparticle friction, were rather negligible [51]. The curves nearly collapsed for the bidisperse MR fluid; nevertheless, noticeable divergence was found at greater Mason numbers, i.e., Mn > 10^−4^. The extent of variance in data related to the type of NPs was low, while the highest manifestation was detected for the MR fluid supplemented with NPs@TEOS (Figure 8b). From this reason, it became obvious that friction forces played a considerable role in the rheology of the bidisperse MR fluids at high values of *Mn*.

As mentioned earlier, a compelling reason for utilizing NP additives in MR fluids is their capacity to improve the sedimentation profile. In order to evaluate such enhancing effects, recorded images were translated into sedimentation curves, as displayed in Figure 9. The reference MR fluid presented the worst sedimentation profile due to the lack of additives acting as physical barriers, in conjunction with possible aggregation of the CI particles [52]. An unexpected behavior was observed for samples containing the neat NPs and NPs@TEOS, respectively. The former gave rise to a superior sedimentation curve, although the neat NPs were presumed to have a higher bulk density compared to their TEOS-coated analogues. This phenomenon could be rationalized by recalling the given surface effects; the neat NPs possessed a rough surface (Figure 1a) while their TEOS-coated analogues had a smooth finish and lower SSA, which diminished friction forces and facilitated sedimentation as a consequence. Another notable aspect was that the NPs@TEOS particles were not attached to the CI particles, instead they remained in the supernatant throughout the sedimentation process (see the digital photo in Figure 9). This confirmed the hypothesis that the NPs@TEOS material acted as a lubricant [46]. The MR suspension supplemented with NPs@m-TEOS showed the best sedimentation profile for the following reasons: (i) it exerted a physical barrier effect; (ii) frictional forces were heightened; (iii) hydrodynamic volume increased; and (iv) its density was lower, thus, it demonstrated greater buoyancy than the counterparts with compact coatings [11,13,53]. These results correlated well with the off-state MR data (Figure 6a). For further analysis, the sedimentation process was simulated by parametric models. In this context, the sedimentation behavior of MR fluids is usually approximated by an exponential decay law, as follows:(7)$$\frac{X\left(t\right)}{X_{0}}=\exp\left(-\frac{t}{t_{\mathrm{rel}}}\right)$$
where *X*(*t*) is the height of the particle-rich phase at time *t*, *X*_0_ represents the total height of the MR fluid and *t*_rel_ denotes the time constant, known as the relaxation time [54]. This equation is only applicable for MR fluids with a narrow particle-size distribution, though it can be modified to address bidisperse/dimorphic MR systems in this form:(8)$$\frac{X\left(t\right)}{X_{0}}=\exp{\left(-\frac{t}{t_{\mathrm{rel}}}\right)}^{\beta }$$
which shares the same parameters defined above, while the *β* exponent (attaining values of 0 < *β* ≤ 1) represents the breadth of the relaxation time distribution [55]. A lower value for the *β* parameter constitutes a wider distribution of relaxation times, whereas Equation (8) collapses into Equation (7) for *β* = 1. Herein, since the expanded model had not provided a satisfactory correlation with the data, a five-parameter (5P) equation was employed, as recently reported by Kang et al. [56]. The 5P model, dubbed the “S-model”, showed an exceptional fitting capacity for describing the sedimentation behavior of kaolinite solutions, and its applicability has been postulated also for other systems. The versatility of the S-model stems from its ability to capture all the characteristic phases of the sedimentation process: (i) hindered sedimentation (a flat top); (ii) constant-speed sedimentation (a measure between two inflexion points); and (iii) a consolidation phase (a slightly sloped tail). The S-model can be written as follows:(9)$$\frac{X\left(t\right)}{X_{0}}=\frac{1}{{\left\{\ln\left[\exp\left(1\right)+{\left(\frac{t}{t_{h}}\right)}^{n_{S}}\right]\right\}}^{m_{C}}}\left\{1-\left[\frac{\ln\left(1+\frac{t}{t_{S}}\right)}{\ln\left(1+\frac{10^{45}}{t_{C}}\right)}\right]\right\}$$
where *t*_h_ is a parameter related to the induction period of hindered sedimentation, *n*_S_ pertains to the steepness of the slope during phase-(ii) sedimentation (see above), m_C_ relates to the shape of the settling curve as it approaches consolidation, *t*_S_ is a parameter related to the slope of hindered settling and *t*_C_ is related to the final height of the sediment [56].

Figure 9a reveals that the S-model fitted well with the MR experimental data (*R*^2^ > 0.980), and Table 2 summarizes the values of its parameters. Generally, the sedimentation tended to occur rather quickly as the oil employed was of low viscosity; this, however, clearly exposed the differences among the samples. Despite relatively fast sedimentation, the MR fluids supplemented with NPs@TEOS and NPs@m-TEOS demonstrated a longer induction period as a consequence of steric hindrance, which was reflected in the higher *t*_h_ values. The reference suspension exhibited the greatest *n*_S_ parameter, reflecting the peak sedimentation rate observed during phase-(ii). The value for *n*_S_ notably decreased after adding the neat NPs and NPs@TEOS. Another significant enhancement was observed in the bidisperse MR fluid with NPs@m-TEOS, which in terms of *n*_S_ values presented a sedimentation rate six times lower than the reference. This finding clearly demonstrated the importance of such additives with a high surface-to-volume ratio. In addition, the *m*_C_ time parameter reflected that the MR fluid supplemented with NPs@m-TEOS required the longest time to approach consolidation. All the MR suspensions exhibited similar *t*_S_ values since differences in the slopes of the “flat top” phase were rather minor. Finally, the values for the *t*_C_ parameter indicated that the bidisperse variant supplemented with NPs@m-TEOS formed the highest sediment, which is a feature associated with facile redispersibility [57]. An analogous situation was observed for the high viscosity oil (100 mPa·s; see Figure 9b), where the MR fluids remained stable in the timeframe of hours.

## 3. Materials and Methods

### 3.1. Materials

The CI particles (SL grade) were supplied by BASF (Ludwigshafen, Germany) and served as the main magnetic constituent. The chemicals utilized to synthesize the NPs and their coatings were obtained from Sigma-Aldrich (St. Louis, MO, USA); if not stated otherwise, these comprised: ferric chloride hexahydrate (≥98%), trisodium citrate dihydrate (≥98%), sodium acetate (anhydrous, ≥99.0%), tetraethoxysilane (TEOS) (98%), cetyltrimethylammonium bromide (CTAB) (≥98%), triethanolamine (≥99.0%), ethylene glycol (anhydrous, 99.8%) and aqueous ammonia solution (ACS, 28–30%). Cyclohexane (ACS, ≥99.5%) was produced by Normapur (USA). Other reagents, namely hydrochloric acid (HCl, 35%), toluene (p.a.) and ethanol (96%), were obtained from Penta Labs (Prague, Czech Republic). Silicone oil (Lukosiol M15 and M100, with the dynamic viscosity of 15 and 100 mPa·s, respectively, and the density of 0.93 g·cm^−3^ at 25 °C) was purchased from Chemical Works Kolín (Czech Republic). Distilled water was used throughout the synthesis process. All the chemicals were employed without further purification, except for trisodium citrate dihydrate which was treated at 160 °C to eliminate water content prior to use. 

### 3.2. Synthesis of the Magnetic NPs

These were fabricated via a simple solvothermal method, similar to that reported by Wei et al. [24]. Ferric chloride hexahydrate, trisodium citrate and ethylene glycol served as the source of iron, the stabilizer and reaction medium, respectively. In a typical procedure, 1.5 g of ferric chloride hexahydrate, 0.36 g of trisodium citrate and 2.4 g of sodium acetate were dissolved in 50 mL of ethylene glycol. The process was accelerated by magnetic stirring at an elevated temperature (35 °C). The resultant mixture was transferred into a Teflon-lined autoclave and sealed. The assembly was placed in a heating oven (200 °C) for 10 h to proceed the reaction. The black precipitate was subsequently collected and washed several times with an excess of water (4 × 50 mL, each) and ethanol (4 × 50 mL, each). The magnet-assisted decantation method was applied, and 0.40 ± 0.03 g of magnetic nano-powder was obtained per synthesis.

### 3.3. TEOS Coating of the NPs

Applying the organosiloxane layer to the NPs was performed as follows: 0.20 g of nano-powder was dispersed in 200 mL of a water/ethanol mixture (at the volume ratio of 1:4), while concurrently adding 3.0 mL of aqueous ammonia solution. Afterwards, 0.40 mL of TEOS was added and the resultant mixture was mechanically agitated at 30 °C for 6 h, giving rise to the NPs@TEOS [24]. The product was repeatedly washed with ethanol (4 × 50 mL, each) in conjunction with magnet-assisted decantation and use of a high-speed centrifuge (Sorvall LYNX 4000, Thermo Scientific, Waltham, MA, USA) operating at 10,000 rpm for 10 min. The powder was dried at 60 °C overnight.

### 3.4. Encapsulation of the NPs@TEOS in Mesoporous Silica

The TEOS-coated NPs were encapsulated in mesoporous silica by applying a method of one-pot biphase stratification [22,24], with CTAB as the surfactant template, resulting in the NPs@m-TEOS product. In this reaction, 0.2 g of NPs@TEOS was dispersed in 30 mL of water, 2.0 g of CTAB and 72 mg of triethanolamine, and the mixture was gently stirred (200 rpm) at 60 °C for 2 h. Subsequently, 10 mL of TEOS dissolved in cyclohexane (10 *v*/*v*%) was carefully added (1 mL/min) into the mixture with the aid of a programmable syringe pump (NE-1000, New Era Pump Systems, Farmingdale, NY, USA), at a constant temperature of 60 °C and a low stirring speed (150 rpm). Although removal of CTAB by calcination is usually possible [24] at 550 °C, in the case of magnetic substrates, such a process causes significant deterioration in magnetic properties. Therefore, CTAB was removed by dispersing the NPs in 120 mL of ethanol that contained 80 mg of ammonium nitrate [22]. The final step of cleaning involved washing in water (4 × 50 mL, each) and ethanol (4 × 50 mL, each) in a similar manner to that described above. The complete procedure for synthesis is illustrated in Figure 10.

### 3.5. General Characterization

The morphology of the CI and NPs was investigated with transmission electron microscopy (TEM) on a JEM-2100Plus instrument (JEOL, Tokyo, Japan), equipped with a lanthanum hexaboride LaB6 emission source, operating at an acceleration voltage of 200 kV. Prior to analysis, the particles were dispersed in ethanol (0.5% *w*/*w*), ultrasonicated (Elmasonic, S10 H, Singen, Germany) and carefully dripped onto a carbon-coated copper grid (300 mesh, Agar Scientific, UK). The liquid phase was evaporated, and the grid was placed on a rod holder for characterization with TEM. The images acquired were analyzed with ImageJ software (version 1.52a, National Institutes of Health, USA) to obtain the particle-size distributions. The statistics presented (average ± standard deviation) were calculated in consideration of at least 300 particles in each specimen.

Scanning electron microscopy (SEM) was performed on a field-emission NanoSEM 450 device (FEI Company, Hillsboro, OR, USA), equipped with a low-vacuum detector, operating at an accelerating voltage of 5 kV (the spot size of 3). Prior to such investigation, a thin layer of the NPs was deposited onto double-sided carbon tape attached to an aluminum stub pin. The SEM device enabled the recording of the energy-dispersive X-ray (EDX) spectra, after employing the Octane Plus detector. 

Fourier-transform infrared spectroscopy (FTIR) data was collected on a Nicolet 6700 device (Thermo Scientific, Waltham, MA, USA) fitted with an ATR module and Ge crystal. The spectra were taken for a wavenumber range of 4000 to 700 cm^−1^, encompassing 64 scans and a spectral resolution of 2 cm^−1^.

Nitrogen absorption/desorption isotherms were recorded with a volumetric gas adsorption analyzer (BELSORP Mini II, BEL, Osaka, Japan) at 77 K. Prior to taking measurements, the samples of powder were degassed in test tubes at 60 °C for 5 h. The specific surface area (SSA) was determined with multi-point Brunauer–Emmett–Teller (BET) analysis, applying at least five data points within a relative pressure range of 0.05 to 0.20 p/p_0_ (Data Analysis Software, version 6.4.1.0, MicrotracBEL Corp, Japan).

Vibrating sample magnetometry (VSM) was carried out to determine the magnetic properties of the CI and NPs on a Model 7404 unit (Lake Shore, Westerville, OH, USA), with external magnetic fields set to ±15 kOe (±1150 kA/m), under laboratory temperature. An amount of the dried powder (20 mg) was placed in a VSM sample holder (730931 Kel-F, powder upper/bottom cup) that was mounted onto a fiberglass tail vibrating at a frequency of 82 Hz and amplitude of 1.5 mm.

X-ray diffraction (XRD) patterns were recorded on a Miniflex 600 (Rigaku, Tokyo, Japan) device with a Co-Kα radiation source (λ = 1.789 Å) across a 2θ range of 20° to 100°, operating at a scan speed of 3°/min.

### 3.6. Preparation of Suspensions and their Magnetorheology

The MR formulations were prepared by dispersing the calculated amount of the CI particles in silicone oil, resulting in the reference MR suspension having a particle concentration of 60 wt%. The bidisperse MR analogues were obtained by supplementing the reference suspension with the additional 2.5 wt% of various sub-micro NPs, i.e., neat NPs, NPs@TEOS and NPs@m-TEOS. The homogenization was ensured by thorough mechanical stirring and subsequent sonication using an ultrasonic processor (UP400S, Hielscher Technologies, Teltow, Germany) equipped with a titanium sonotrode (H7). 

The rheological behavior of the tested MR formulations was investigated on a Physica rotational rheometer (MCR502, Anton Paar, Graz, Austria) equipped with a plate–plate accessory (PP20/MRD/TI). The lower plate was composed of steel (DIN 1.0718–11SMnPb30), while the upper one was constructed from non-magnetic titanium (DIN 3.7165–Ti 6Al 4V). The configuration was characterized by the plate diameter of 20 mm, gapped 0.3 mm from the lower one, which required 0.10 mL of the MR fluid that was dosed with a micropipette. A homogeneous magnetic field was generated with the magneto-cell module (MRD170/1T), and a true magnetic field strength (up to 360 kA/m at increments of 72 kA/m) was verified on a teslameter (Magnet Physik, FH51, Dr. Steingroever, Köln, Germany). 

The testing procedure comprised the following steps: (i) the given sample was pre-sheared in the off-state under a constant shear rate of 100 s^−1^ for a duration of one minute; pre-shearing in this way eliminated the rheological history of each sample and imposed the same initial conditions; (ii) the desired magnetic field was applied under static conditions for a period of one minute to provide sufficient time for particle-chain development; (iii) in the case of shear experiments, rheological data was collected within the range of shear rates of 10^−1^ to 5 × 10^3^ s^−1^. The repeatability of the particle structure formation was investigated under a dynamic magnetic field that was periodically switched on/off at intervals of 30 s. The conditions for this test included a constant shear rate of 50 s^−1^ and magnetic field of 216 kA/m. A thermostatic device (Julabo FS18, Seelbach, Germany) was employed to maintain the desired temperature (25 °C) throughout the period of taking measurements. Each sample was tested three times to ensure reproducibility of the data; mathematical modelling was performed with mean average values.

### 3.7. Sedimentation Stability

The homogenized bidisperse MR fluids were added into measuring cuvettes and their sedimentation ratio, expressed as the height of the particle-rich fraction relative to the initial height of the suspension, was monitored on a digital camera (DMK 42BUC03, ImagingSource, Bremen, Germany) equipped with a manual iris varifocal lens (T3Z3510CS, Computar, Japan). The images obtained were analyzed with ImageJ software (version 1.52a, National Institute of Health, USA).

## 4. Conclusions

MR fluids are remarkable materials with a rich history; however, their sedimentation instability is still a limiting factor for their practical utilization. Herein, we presented a new concept that employed bidispersal and coating techniques to markedly improve their sedimentation profiles, while also exerting a minimal impact on MR performance. The sub-micro additives were fabricated by solvothermal synthesis and further encapsulated either with a compact or mesoporous-TEOS coating via the biphase stratification method. The quality of the coatings was investigated with TEM, SEM and BET analysis; this revealed that the surfaces of the neat NPs were rough, whereas the NPs@TEOS particles were smooth, and the NPs@m-TEOS particles possessed radially oriented pores, their respective SSA values being 47, 9.5 and 312 m^2^/g. The NP additives were free of impurities, as proven with EDX and FTIR. Their magnetic properties were fitted via the Jiles–Atherton equation, giving *M*_S_ values of 55.6, 34.1 and 23.7 emu/g, respectively, which correlated well with the (m-)TEOS thickness layer. The bidisperse MR suspensions exhibited non-trivial behavior; the presence of the NPs prevented agglomeration of the primary CI particles, reflected in the slightly lower off-state viscosity. In the on-state, they demonstrated a comparable or slightly higher performance than the reference, caused by collective magnetization, while a slight decrease was observed under moderate to high magnetic fields (above 144 kA/m). Likewise, the dynamic yield stress as predicted by the modified Cross model followed a similar trend. Mason numbers were used to produce single master curves, and the results showed that magnetostatic and hydrodynamic forces dominated in the reference MR fluids, while other forces, most likely the friction, played a certain role in the bidisperse MR analogues. Sedimentation behavior was, for the first time in magnetorheology, fitted with the “S-model” with excellent accuracy and physical meaning of all the parameters. Bidispersal proved to be an effective strategy for reducing sedimentation, since the MR fluid supplemented with NPs@m-TEOS exhibited a sedimentation rate six times lower than the reference. Future research could explore the fabrication of more complex MR fluids having multiple constituents in their composition. 

## Figures and Tables

**Figure 1 ijms-23-11044-f001:**
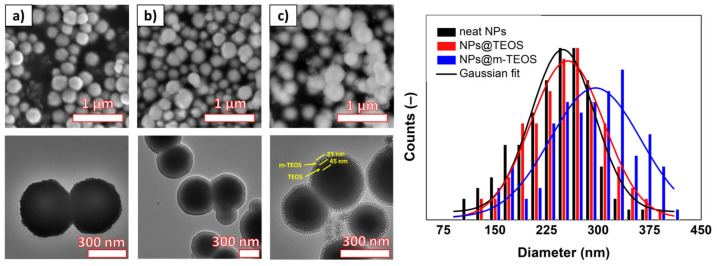
SEM/TEM images of the (**a**) neat NPs, (**b**) NPs@TEOS and (**c**) NPs@m-TEOS, and the particle-size distributions for the corresponding nano-species.

**Figure 2 ijms-23-11044-f002:**
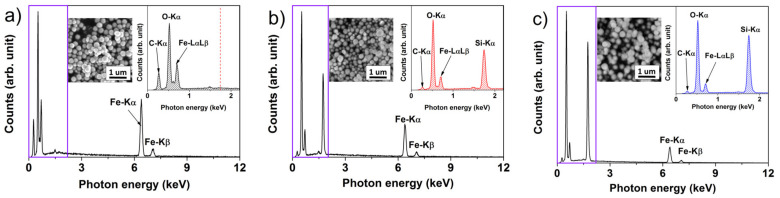
EDX spectra for the (**a**) neat NPs, (**b**) NPs@TEOS and (**c**) NPs@m-TEOS; these spectra were recorded from SEM insets (from an area of 2.5 × 2.5 µm^2^); the dashed line in (**a**) represents the position of Si-Kα photon energy absent from the spectrum.

**Figure 3 ijms-23-11044-f003:**
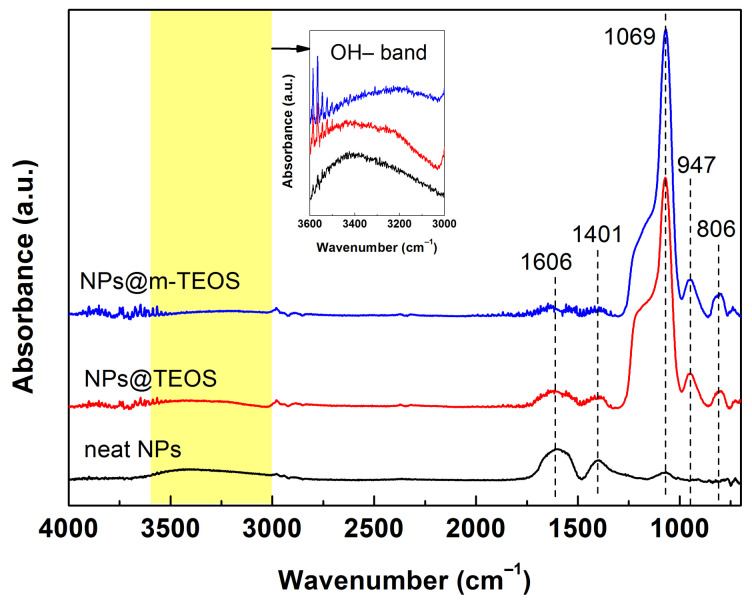
FTIR spectra for the neat NPs, NPs@TEOS and NPs@m-TEOS particles with denoted characteristic wavenumbers.

**Figure 4 ijms-23-11044-f004:**
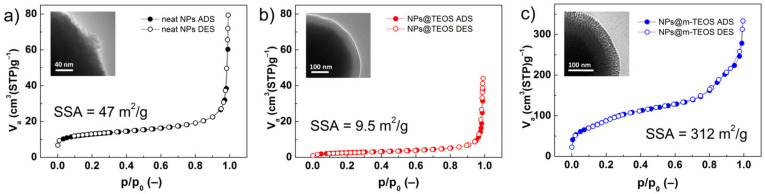
Nitrogen adsorption/desorption (*solid/open symbols*) isotherms for the (**a**) neat NPs, (**b**) NPs@TEOS and (**c**) NPs@m-TEOS; the insets highlight differences in surface topology.

**Figure 5 ijms-23-11044-f005:**
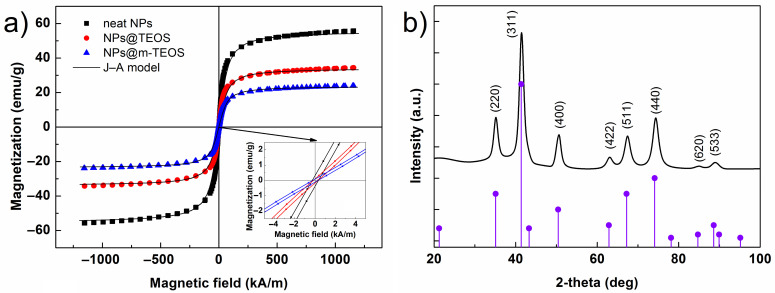
Isothermal VSM curves for the (**a**) neat NPs, NPs@TEOS and NPs@m-TEOS particles; the inset shows a magnified view of the zero-field region; graph (**b**) describes the XRD spectrum for the neat NPs, with vertical bars representing the Bragg positions.

**Figure 6 ijms-23-11044-f006:**
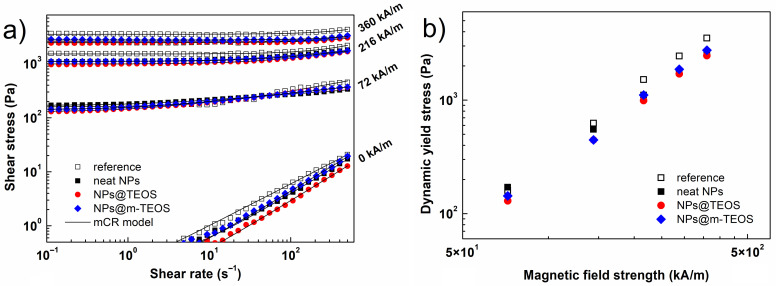
Shear stress as a function of shear rate (**a**), and the dynamic yield stress as a function of magnetic field strength (**b**) for the reference MR fluid (white squares), and its bidisperse analogues supplemented with the neat NPs (black squares), NPs@TEOS (red circles) and NPs@m-TEOS (blue diamonds).

**Figure 7 ijms-23-11044-f007:**
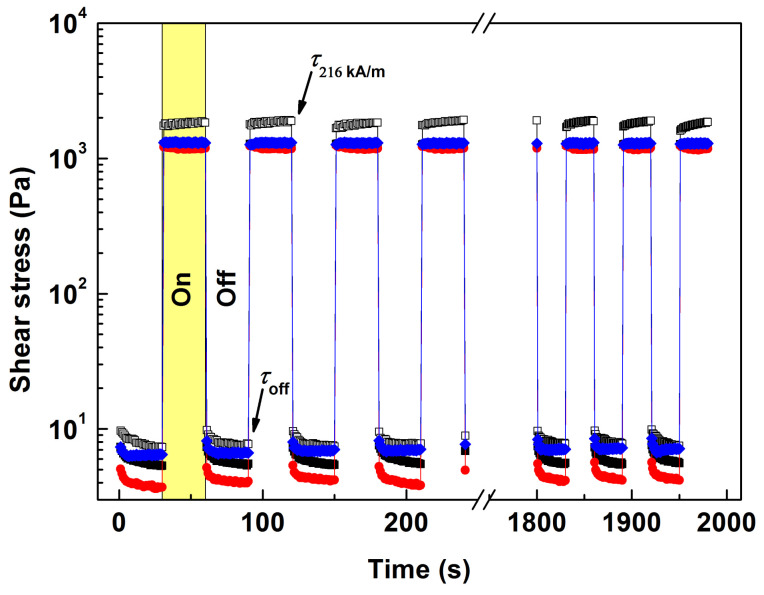
Shear stress response under a dynamic magnetic field (216 kA/m) for the reference MR fluid (white squares) and its bidisperse analogues supplemented with the neat NPs (black squares), NPs@TEOS (red circles) and NPs@m-TEOS (blue diamonds).

**Figure 8 ijms-23-11044-f008:**
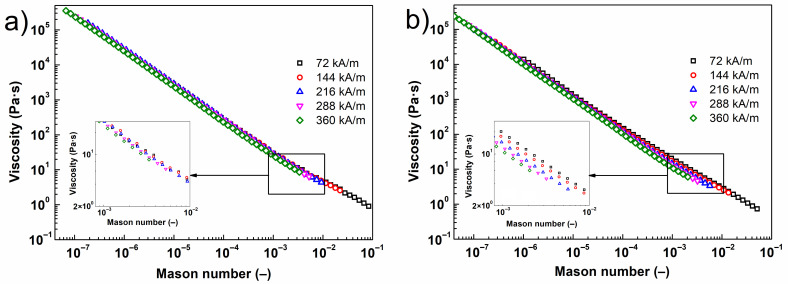
Shear viscosity as a function of Mason number for the reference MR fluid (**a**) and its bidisperse analogues supplemented with NPs@TEOS (**b**) under various strengths of magnetic field.

**Figure 9 ijms-23-11044-f009:**
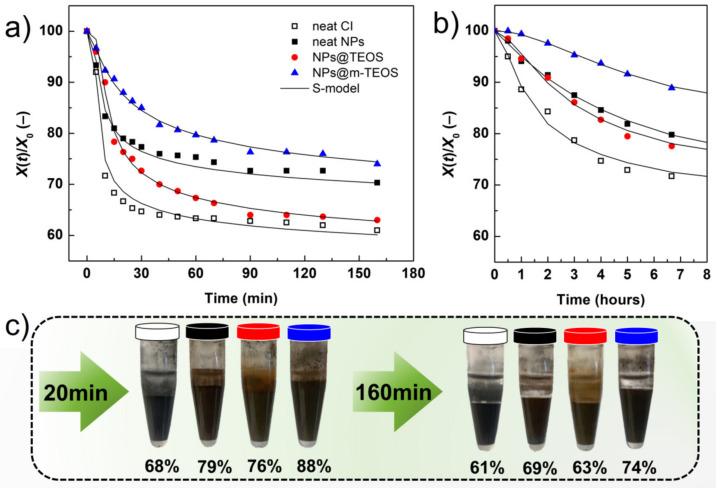
Sedimentation curves for the reference MR fluid (white squares) and its bidisperse analogues supplemented with the neat NPs (black squares), NPs@TEOS (red circles) and NPs@m-TEOS (blue triangles) in (**a**) low viscosity (15 mPa·s) oil and (**b**) high viscosity (100 mPa·s) oil; the solid lines mark the best fit of the 5P S-model. The section (**c**) shows the digital images of the MR fluids based on low viscosity oil at two different sedimentation periods. The color of each stopper denotes the type of the sample, as per legend in (**a**) section.

**Figure 10 ijms-23-11044-f010:**
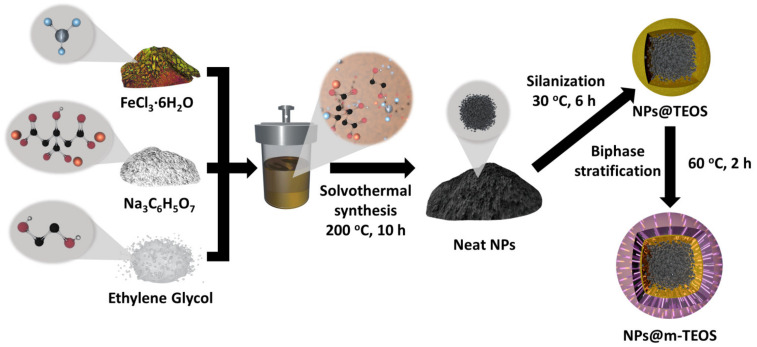
Reaction scheme comprising solvothermal synthesis, silanization and biphase stratification.

**Table 1 ijms-23-11044-t001:** Overview of nano-fillers and related SSA values, as documented in recent research on MR fluids.

Sample ID	Coating Material	Fabrication Technique	Dosing in MR Fluid	Diameter	*M*_S_ (emu/g)	SSA (m^2^/g)	Literature
Hydrophilic silica	N/D	Commercial	0.5 wt%	7 nm	N/D	395	Felicia et al. [34] (2015)
Iron 1,3,5-benzenetricarboxylate	N/D	Commercial	1.0 wt%	Agglomerate	0.185	487	Quan et al. [39] (2015)
Flower-like Fe_3_O_4_	N/D	Solvothermal method	Not specified	3.2 µm	58.8	62.78	Wang et al. [38] (2019)
Fe_3_O_4_	N/D	Co-precipitation	25 vol%	10–15 nm	68.2	81.15	Du et al. [30] (2020)
Fe_3_O_4_ embedded in carbon layer	Carbon	Hydrothermal method and calcinations	25–35 vol%	150–200 nm	38.3	186.49	
Fe_3_O_4_ embedded in dense carbon layer	Carbon				32.0	178.59	
MnFe_2_O_4_/MgAl-LDH	Hybrid	Two-step hydrothermal method	25 wt%	>44 nm	33.4	119.04	He et al. [36] (2020)
MnFe_2_O_4_ flakes	N/D	One-step solvothermal method	Not specified	Several microns (lateral size), thickness of 21 nm	58.8	92.86	Wang et al. [40] (2020)
Hydrophobic silica	N/D	Commercial	3.0 wt%	40–300 nm	N/D	115–125	Aruna et al. [32] (2021)
Hydrophilic silica				0.2–3 µm		135–200	
Polypyrrole-coated magnetite	Hybrids	One-pot oxidation	5.0 wt%	Agglomerate	3–51	40.4–58.1	Stejskal et al. [37] (2021)
NPs@m-TEOS	Mesoporous TEOS	Biphase stratification	3.0 wt%	296 ± 51 nm	23.7 *	312	This paper

* investigated below.

**Table 2 ijms-23-11044-t002:** Numerical fitting results for the 5P S-model.

Sample ID	*t* _h_	*n* _S_	*m* _C_	*t* _S_	*t* _C_
Reference	4.26	8.96	0.14	50.1	0.031
Neat NPs	4.16	4.35	0.12	50.1	0.002
NPs@TEOS	6.81	4.28	0.17	50.0	0.876
NPs@m-TEOS	6.72	1.44	0.18	50.0	0.884

## Data Availability

Not applicable.

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
