# Peer review of "Stable Magnetorheological Fluids Containing Bidisperse Fillers with Compact/Mesoporous Silica Coatings"

_ijms, 2022, doi:10.3390/ijms231911044_

Round 1

Reviewer 1 Report

This paper enhances the stability of MRF by using bidisperse fillers with compact silica coatings. It is a well written paper with appropriate investigations that leads to good results and discussion. Only minor concerns should be looked at especially on the reduction of the Ms from 55.6 to 23.7 emu/g (Figure 6(a)), more than 50%. However,  with referring to Figure 8, NPs@TEOS has the lowest shear stress at off-state and approximately the same with NPs@m-TEOS at on-state. It shows that the absolute MR effect of NPs@TEOS is the highest at magnetic field of 216 kA/m. Since, it is not clear why the word “bidisperse” is used or in other words, the preparation of MRF is not clearly stated. The amount or weight % of magnetic particles in MRF plays an important role in determining the MR effect in which is the main concern of utilizing MR fluid in any devices.

Other than that, Table 1 is rather to be omitted or places in introduction section. 

Reviewer 2 Report

Dear Authors. The paper is well-prepared. Therefore, I will agree to the publication after some modifications:

Q1: Remove all the abbreviations from the abstract.

Q2: Line 56: Numbers in the chemical formula Fe3O4 should be subscript

Q3: Figure 7: What does “OFF” mean, and why the part of the curves are not visible?

Q4: I think it should be a break between the given number and standard deviation in the whole manuscript (for instance, in line 384)

Q5: The list of references is too long. Please remove some, especially when you put a few for the same text in the manuscript ( for instance, in line 290) . I do not understand how you access publication no 49, which is not yet published. Did you consider the following papers:

Effect of temperature on sedimentation stability and flow characteristics of magnetorheological fluids with damper as the performance analyzer.

Magnetorheological fluid based on submicrometric silica-coated magnetite particles under an oscillatory magnetic field

The effect of nano-silica and nano-magnetite on the magnetorheological fluid stabilization and magnetorheological effect
